# Cosmic-void observations reconciled with primordial magnetogenesis

**David N. Hosking** [1,2,3,4] ✉ **& Alexander A. Schekochihin**[3,5]

It has been suggested that the weak magnetic field hosted by the intergalactic medium in cosmic voids could be a relic from the early Universe. However, accepted models of turbulent magnetohydrodynamic decay predict that the present-day strength of fields originally generated at the electroweak phase transition (EWPT) without parity violation would be too low to explain the observed scattering of $\gamma$-rays from TeV blazars. Here, we propose that the decay is mediated by magnetic reconnection and conserves the mean square fluctuation level of magnetic helicity. We find that the relic fields would be stronger by several orders of magnitude under this theory than was indicated by previous treatments, which restores the consistency of the EWPT-relic hypothesis with the observational constraints. Moreover, efficient EWPT magnetogenesis would produce relics at the strength required to resolve the Hubble tension via magnetic effects at recombination and seed galaxy-cluster fields close to their present-day strength.

It is widely believed that cosmic voids host magnetic fields. Evidence for this comes chiefly from $\gamma$-ray observations of blazars[1–12] (see refs. 13–15 for reviews): extragalactic magnetic fields (EGMFs) in voids would, if present, scatter the electrons produced in electromagnetic cascades of TeV $\gamma$-rays emitted by blazars, thus suppressing the number of secondary (GeV) $\gamma$-rays received at Earth. Such suppression is indeed observed, and can be used to constrain the root mean square strength $B \equiv \langle \mathbf{B}^2 \rangle^{1/2}$ and energy-containing scale $\lambda_B$ of the magnetic fields. Using spectra measured by the *Fermi* telescope, refs. 3,12 estimate that

$$B \gtrsim 10^{-17}\,\text{G} \left( \frac{\lambda_B}{1\text{Mpc}} \right)^{-1/2}, \qquad (1)$$

where $10^{-17}\,\text{G}$ can increase to $10^{-15}\,\text{G}$ depending on modelling assumptions, including the effect of time delay due to the larger distance travelled by scattered electrons[3,16]. Equation (1) may also be subject to some modification due to the cooling of cascade electrons by plasma instabilities[17–20]—what effect, if any, this has on the constraint (1) is poorly understood—see refs. 21,22 for recent discussions.

Where might fields in voids come from? A popular idea (although not the only one, see ref. 23) is that they could be relics of primordial magnetic fields (PMFs) generated in the early Universe[24], including, prominently, at the electroweak phase transition (EWPT)[25]. If so, the physics of the early Universe could be constrained by observations of the fields in voids—a remarkable possibility—provided the magnetohydrodynamic (MHD) decay of the PMFs between their genesis and the present day were understood. However, the conventional theory of the decay[24] (see refs. 13–15 for reviews) appears inconsistent with the EWPT-relic hypothesis: ref. 26 argue that the lower bound (1) on $B$ is too high to be consistent with PMFs generated at the EWPT without magnetic helicity (a topological quantity that quantifies the number of twists and linkages in the field, which is conserved even as energy decays[27]). Furthermore, they show that the amount of magnetic helicity required for consistency with Equation (1) is greater than can be generated by baryon asymmetry at the EWPT, as estimated by ref. 28. In principle, other mechanisms of magnetic-helicity generation may have been present in the early Universe; one idea is chiral MHD (see[29] and references therein). Whether enough net helicity can be generated via these mechanisms for PMFs to become

[1]Oxford Astrophysics, Denys Wilkinson Building, Keble Road, Oxford OX1 3RH, UK. [2]Princeton Center for Theoretical Science, Princeton University, Princeton, NJ 08544, USA. [3]Merton College, Merton Street, Oxford OX1 4JD, UK. [4]Gonville & Caius College, Trinity Street, Cambridge CB2 1TA, UK. [5]The Rudolf Peierls Centre for Theoretical Physics, University of Oxford, Clarendon Laboratory, Parks Road, Oxford OX1 3PU, UK. ✉e-mail: dhosking@princeton.edu

maximally helical during their evolution remains an open question[30,31]. On the other hand, ref. 26 note that their conclusions could be subject to modification by the contemporaneous discovery of the inverse transfer of magnetic energy in simulations of non-helical MHD turbulence[32,33] (see refs. 30,34−36 for schemes for modifying their conclusions based on decay laws obtained numerically). The inverse transfer was discovered by ref. 37 to be a consequence of local fluctuations in the magnetic helicity, which are generically present even when the global helicity vanishes, and whose mean square fluctuation level is conserved.

In this paper, we apply the theory of ref. 37 to the problem of predicting the strength of the relics of PMFs. We find that the constraint imposed by magnetic-helicity conservation, when taken together with the other key result of ref. 37, and of refs. 38−40, that the decay timescale is the one on which magnetic fields reconnect, restores consistency of the hypothesis of a non-helical EWPT-generated PMF with Equation (1). We also find that reasonably efficient magnetogenesis of non-helical magnetic field at the EWPT could produce relics with around the $10^{-11}$ G comoving strength that, it has been suggested, is sufficient to resolve the Hubble tension[41,42]. Relics of this strength would also constitute seed fields for galaxy clusters that would not require much amplification by turbulent dynamo after structure formation to reach their observed present-day strength[43] (although dynamo would still be required to maintain cluster fields at present levels).

## Results

We take the metric of the expanding Universe to be

$$ds^2 = a^2(t)(-dt^2 + dx_i\, dx^i), \qquad (2)$$

where $a(t)$ is the scale factor, normalised to 1 at the present day, $t$ is conformal time (related to cosmic time $\bar{t}$ by $a(t)dt = d\bar{t}$), and $x_i$ are comoving coordinates. The expanding Universe MHD equations can be transformed to those for a static Universe by a simple rescaling[44]: the scaled variables

$$\tilde{\rho} = a^4\rho, \qquad \tilde{p} = a^4 p, \quad \tilde{\mathbf{B}} = a^2\mathbf{B}, \quad \tilde{\mathbf{u}} = \mathbf{u}, \\ \tilde{\eta} = \eta/a, \quad \tilde{\nu} = \nu/a, \qquad (3)$$

[where $\rho$, $p$, **B, u,** $\eta$, and $\nu$ are the physical values of the total (matter + radiation) density, pressure, magnetic field, velocity, magnetic diffusivity, and kinematic viscosity, respectively] evolve according to the MHD equations in Minkowski spacetime. As in previous work (see refs. 13−15), we consider the dynamics of the tilded variables in Minkowski spacetime and transform the result to the spacetime (2) of the expanding Universe via Equation (3).

### Selective decay of small-scale structure

Historically, it has been believed that statistically isotropic MHD turbulence decays while preserving the small-$k$ asymptotic of the magnetic-energy spectrum $\mathcal{E}_M(k)$ (see refs. 13,14 and references therein). This idea, sometimes called selective decay of small-scale structure, amounts to a statement of the invariance in time of the magnetic Loitsyansky integral,

$$I_{\mathbf{L}_M} \equiv -\int d^3\mathbf{r}\, r^2 \langle \tilde{\mathbf{B}}(\mathbf{x})\tilde{\mathbf{B}}(\mathbf{x}+\mathbf{r})\rangle, \qquad (4)$$

where angle brackets denote an ensemble average. For isotropic turbulence without long-range spatial correlations, $I_{\mathbf{L}_M}$ is related to $\mathcal{E}_M(k)$ by

$$\mathcal{E}_M(k \to 0) = \frac{I_{\mathbf{L}_M} k^4}{24\pi^2} + O(k^6). \qquad (5)$$

Invariance of $I_{\mathbf{L}_M}$ implies

$$I_{\mathbf{L}_M} \sim \tilde{B}^2 \lambda_B^5 \sim \text{const.} \qquad (6)$$

Here and in what follows, we use the symbol ~ to denote equality up to a dimensionless number of order unity. In writing Equation (6), we have assumed that the magnetic-energy spectrum is sufficiently peaked around the energy-containing scale $\lambda_B$ for the latter to be equal to the correlation, or integral, scale of the field. This would not be the case for a scale-invariant magnetic field (often conjectured to be generated by inflationary mechanisms). We exclude such fields from our analysis in this paper, in which we consider causal fields−the sort that could be generated at a phase transition−exclusively.

### Decay timescale

Equation (6) can be translated into a decay law for magnetic energy by a suitable assumption about how the energy-decay timescale,

$$\tau(\tilde{B},\lambda_B,t) \equiv -\left(\frac{d\log\tilde{B}^2}{dt}\right)^{-1}, \qquad (7)$$

depends on $\tilde{B}$, $\lambda_B$ and $t$. Regardless of this choice, Equations (6) and (7) have the following important property. Suppose that, after some intermediate time $t_c$, $\tau(B,\lambda_B,t)$ can be approximated by some particular product of powers of its arguments. Then, for all $t \gg \tau(t_c)$, $\tilde{B}^2$ decays as a power law: $\tilde{B}^2 \propto t^{-p}$, where $p$ is a number of order unity. Substituting this back into Equation (7), one finds

$$\tau(\tilde{B},\lambda_B,t) \sim t, \qquad (8)$$

which is an implicit equation for $\tilde{B} = \tilde{B}(\lambda_B)$ that can be solved simultaneously with Equation (6) for $\tilde{B}(t)$ and $\lambda_B(t)$. Equation (8) was first suggested by ref. 24 on phenomenological grounds. Its great utility, which has perhaps not been spelled out explicitly, is that it implies that one need not know the functional form of $\tau(\tilde{B},\lambda_B,t)$ during the early stages of the decay in order to compute $\tilde{B}$ and $\lambda_B$ at later times. Thus, the effect of early Universe physics (e.g., neutrino viscosity) on the decay dynamics can be safely neglected.

### Inconsistency with observations

Assuming that the decay satisfies Equation (6) and that its timescale is Alfvénic, viz.,

$$\tau \sim \frac{\lambda_B}{\tilde{v}_A}, \quad \tilde{v}_A = \frac{\tilde{B}}{\sqrt{4\pi\tilde{\rho}_b}}, \qquad (9)$$

when it terminates at the recombination time $t_{\text{recomb}}$[14] [Equation (27) in Methods], Equation (8) implies[24]

$$\tilde{B}(t_{\text{recomb}}) \sim 10^{-8.5}\text{G}\,\frac{\lambda_B(t_{\text{recomb}})}{1\,\text{Mpc}} \qquad (10)$$

[see Equation (31) in Methods]. In (9), $\tilde{\rho}_b$ is the baryon density, which appears because photons do not contribute to the fluid inertia at scale $\lambda_B$ at the time of recombination[45] [see Equation (29) in Methods]. An approximate upper bound, $I_{\mathbf{L}_M,\text{max}}$, on $I_{\mathbf{L}_M}$ follows from assuming that the magnetic-energy density $\tilde{\rho}_B \equiv \tilde{B}^2/8\pi$ and the electromagnetic-radiation density $\tilde{\rho}_\gamma$ were equal at the time $t_*$ of the EWPT while $\lambda_B(t_*)$ was equal to the Hubble radius $r_H(t_*)$. This corresponds to $\tilde{B}(t_*) \sim 10^{-5.5}$ G and $\lambda_B(t_*) \sim r_H(t_*) \sim 10^{-10}$ Mpc[13,26]. As is shown in Fig. 1, these values and Equation (10) together lead to values of $\tilde{B}$ and $\lambda_B$ at $t_{\text{recomb}}$ that violate the observational constraint (1). Note that $\lambda_B(t_*) \sim 10^{-2} r_H(t_*)$ is, in fact, a more popular estimate, corresponding

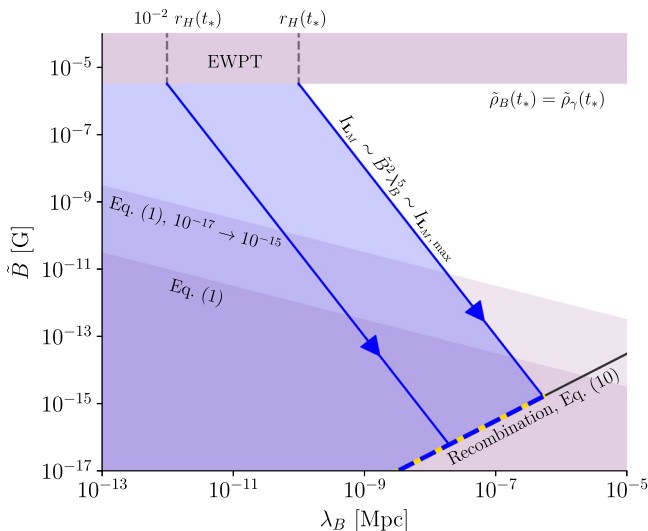

**Fig. 1 | Inconsistency of the decay theory based on Equations (6) and (9) with observational constraints for EWPT-generated PMFs.** Purple regions denote values of $\tilde{B}$ and $\lambda_B$ excluded on physical [$\tilde{\rho}_B(t) \lesssim \tilde{\rho}_\gamma(t_*)$] or observational [the two forms of the constraint (1)] grounds. Under decays that conserve $I_{L_M}$ [Equation (4)], $\tilde{B}$ and $\lambda_B$ evolve along lines parallel to the ones shown in blue. The predicted values of modern-day $\tilde{B}$ and $\lambda_B$ are given by the intersection of these lines with Equation (10). We see that even PMFs generated with $\tilde{\rho}_B(t_*) \sim \tilde{\rho}_\gamma(t_*)$ and $\lambda_B(t_*) \sim r_H(t_*)$ produce modern-day relics that are inconsistent with Equation (1).

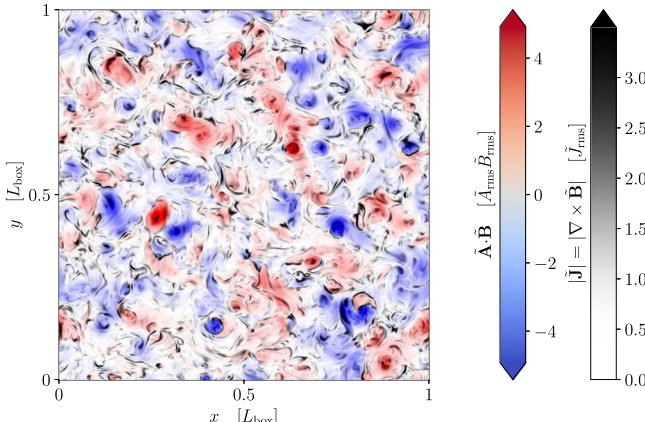

**Fig. 2 | Slice of magnetic-helicity density from a simulation of decaying non-helical MHD turbulence.** The turbulence breaks up into patches of positive and negative helicity $h$ (computed in the Coulomb gauge; $\nabla \tilde{A} = 0$), shown in red and blue, respectively (in units of the product of the root-mean-square values of $\tilde{A}$ and $\tilde{B}$, denoted $\tilde{A}_{rms}$ and $\tilde{B}_{rms}$, respectively). The invariance of $I_H$ [Equation (11)] is a manifestation of the conservation of the net magnetic-helicity fluctuation level arising in large volumes. Because of the complex magnetic-field topology, the rate-setting process for the decay is magnetic reconnection: reconnection sites, indicated in the figure by patches of large current density $|\tilde{\mathbf{J}}| = |\nabla \times \tilde{\mathbf{B}}|$ (black; plotted with a variable-opacity scale in units of the root-mean-square current density, $J_{rms}$), typically form between the helical structures. See the Numerical Simulation section of Methods for details of the numerical setup.

to the typical coalescence size of bubbles of new phase that form at the phase transition[46]; for this initial correlation scale, the predicted value of $\tilde{B}$ is separated from the allowed values by around three orders of magnitude. A similar calculation led ref. 26 to conclude that genesis of EGMFs at the EWPT was unlikely (although we note that significant modification of Equation (1) by inclusion of the effects of plasma instabilities in the modelling of the electromagnetic cascade—see the comment below Equation (1)—could alter this conclusion).

### Saffman helicity invariant

We argue that the theory outlined above requires revision. First, the idea of selective decay of small-scale structure is flawed. This is because the $k\lambda_B \ll 1$ tail of the magnetic-energy spectrum $\mathcal{E}_M(k)$ corresponds not to physical structures (as in the Richardson-cascade picture of inertial-range hydrodynamic turbulence) but to cumulative statistical properties of the structures of size $\lambda_B$[47]. Absent a physical principle to support the invariance of $I_{L_M}$ (such as angular-momentum conservation for its hydrodynamic equivalent[47,48]), there is, therefore, no reason to suppose that the small-$k$ asymptotic of $\mathcal{E}_M(k)$ evolves on a longer timescale than the dynamical one of $\lambda_B$-scale structures (if this is long compared to the magnetic-diffusion timescale at scale $\lambda_B$, then selective decay is valid, as the simulations of[24,49] confirm, but this is not the regime relevant to PMFs).

Instead, we propose that the decay of PMFs is controlled by a different integral invariant[37]:

$$I_H = \int d^3\mathbf{r} \langle h(\mathbf{x})h(\mathbf{x}+\mathbf{r})\rangle, \tag{11}$$

where $h = \tilde{\mathbf{A}}\tilde{\mathbf{B}}$ is the helicity density ($\tilde{\mathbf{B}} = \nabla \times \tilde{\mathbf{A}}$). Equation (11) is equivalent to

$$I_H = \lim_{V\to\infty}\frac{1}{V}\left\langle\left[\int_V d^3\mathbf{x}\,h(\mathbf{x})\right]^2\right\rangle = \lim_{V\to\infty}\frac{\langle H_V^2\rangle}{V}, \tag{12}$$

where $H_V$ is the total magnetic helicity contained within the control volume $V$. The invariance of $I_H$ can therefore be understood intuitively

as expressing the conservation of the net mean square fluctuation level of magnetic helicity per unit volume that arises in any finite volume of non-helical MHD turbulence (see Fig. 2; we refer the reader concerned about the existence of such fluctuations to Section B of the Supplementary Information). Numerical evidence supporting the invariance of $I_H$ has been presented by ref. 37 and independently by refs. 50–52. From $I_H$ = const, we deduce

$$I_H \sim \tilde{B}^4\lambda_B^5 \sim \text{const}. \tag{13}$$

We make two brief remarks. First, growth of $I_{L_M}$, and, therefore, the inverse transfer effect discovered by refs. 32,33,53, follows immediately from Equation (13). This is because $I_{L_M} \sim \tilde{B}^2\lambda_B^5 \sim I_H/\tilde{B}^2$ under self-similar evolution, so that $\mathcal{E}_M(k \to 0) \propto I_{L_M}k^4$ [see Equation (5)] grows while $\tilde{B}$ decays. Second, the value of the large-scale spectral exponent does not affect the late-time limit of the decay laws in our theory (see Section C of the Supplementary Information), unlike in the selective-decay paradigm.

### Reconnection-controlled decay timescale

The second revision that we propose to the existing theory is that the field's decay timescale $\tau$ should be identified not with the Alfvénic timescale (9), but with the magnetic-reconnection one. This is because relaxation of stochastic magnetic fields via the generation of Alfvénic motions is prohibited by topological constraints, which can only be broken by reconnection. Refs. 37,40,50 have presented numerical evidence for a reconnection-controlled timescale for decays that occur with a dominance of magnetic over kinetic energy (see refs. 38,39 for the same in 2D). Magnetically dominated conditions are relevant to the decay of PMFs because (i) the large neutrino and photon viscosities in the early Universe favour them, and (ii) once established, they are maintained, as reconnection is typically slow compared with the Alfvénic timescale. The identification of $\tau$ as the reconnection timescale implies that a number of different decay regimes are possible, as we now explain.

Under resistive-MHD theory, reconnecting structures in a fluid with large conductivity generate a hierarchy of current sheets at increasingly small scales via the plasmoid instability[54]. The global reconnection timescale is the one associated with the smallest of these sheets (the so-called critical sheet), which is short enough to be marginally stable[55,56] (see ref. 57 for a review). This timescale is

$$\tau_{\rm rec} = (1+{\rm Pm})^{1/2} \, \min\left\{ S^{1/2}, S_c^{1/2} \right\} \frac{\lambda_B}{\tilde{v}_A}, \tag{14}$$

where $\mathrm{Pm} = \tilde{\nu}/\tilde{\eta}$ is the magnetic Prandtl number, which appears because viscosity can suppress the outflows that advect reconnected field away from the reconnection site,

$$S = \frac{\tilde{v}_A \lambda_B}{\tilde{\eta}(1+{\rm Pm})^{1/2}} \tag{15}$$

is the Lundquist number based on the reconnection outflow and $S_c \sim 10^4$ is the critical value of $S$ for the onset of the plasmoid instability. Equation (14) is a straightforward theoretical generalisation[57] to arbitrary Pm of a prediction for $\mathrm{Pm}=1$[55] that has been confirmed numerically[56,58]. Pm is given by Spitzer's theory[59] [$\mathrm{Pm_{Sp}} \sim 10^7$ at recombination, see Equation (37) in Methods] if the plasma is collisional, i.e., if the Larmor radius of protons $r_L = m_i c v_{\mathrm{th},i}/aeB$ is large compared to their mean free path, $\lambda_{\rm mfp}$ ($m_i$ and $v_{\mathrm{th},i} \equiv \sqrt{2T/m_i}$ are the mass and thermal speed of protons respectively). If, on the other hand, $r_L < \lambda_{\rm mfp}$, which happens if $B > B_{\rm iso} \equiv m_i c v_{\mathrm{th},i}/ea\lambda_{\rm mfp}$, then the components of the viscosity tensor perpendicular to the magnetic field are reduced by a factor $(r_L/\lambda_{\rm mfp})^2$, because protons' motions across $\tilde{\mathbf{B}}$ are inhibited by their Larmor gyration[60]. These are the components that limit reconnection outflows because velocity gradients in reconnection sheets are perpendicular to the mean magnetic field. Therefore, $\mathrm{Pm} \to (r_L/\lambda_{\rm mfp})^2 \mathrm{Pm_{Sp}} = (\tilde{B}_{\rm iso}/\tilde{B})^2 \mathrm{Pm_{Sp}}$ in Equation (14) if $\tilde{B} > \tilde{B}_{\rm iso} \equiv a^2 B_{\rm iso}$.

The validity of the resistive-MHD treatment that leads to Equation (14) requires the fluid approximation to hold at the scale of the critical sheet: its width

$$\delta_c \sim \frac{S_c^{1/2}}{S} \lambda_B, \tag{16}$$

must be larger than either $r_L$ or the ion inertial length $d_i = \sqrt{m_i c^2 / 4\pi e^2 n_i a^2}$ ($n_i$ is the proton number density)[55,61]. If $\delta_c < r_L, d_i$, then the physics of the critical sheet is kinetic, not fluid, and the reconnection timescale is

$$\tau_{\rm rec} \sim 10 \frac{\lambda_B}{\tilde{v}_A}, \tag{17}$$

rather than (14). Equation (17) is a robust numerical result whose theoretical explanation is an active research topic (see ref. 62 for a recent study,[63,64] for reviews). We shall find in the next section that (17) is not the limiting timescale at recombination for almost any choice of initial condition consistent with EWPT magnetogenesis; our conclusions therefore do not depend sensitively on the validity of (17).

The decay timescale can also be limited by radiation drag due to photons[24]; this imparts a force $-\tilde{\alpha}\tilde{\mathbf{u}}$ per unit density of fluid [see Equation (54) in Methods]. The drag is subdominant to magnetic tension at sufficiently small scales (as it does not depend on gradients of $\tilde{\mathbf{u}}$), so does not contribute to Pm in Equation (14). However, it can inhibit inflows to the reconnection layer. Balancing drag with magnetic tension at the integral scale $\lambda_B$, we find an inflow speed $\tilde{u} \sim \tilde{v}_A^2/\tilde{\alpha}\lambda_B$, so

the timescale for magnetic flux to be processed by reconnection is

$$\tau_\alpha \equiv \frac{\tilde{\alpha}\lambda_B^2}{\tilde{v}_A^2}. \tag{18}$$

The timescale for energy decay depends on whether large-scale drag or small-scale reconnection physics is most restrictive:

$$\tau = \max\{\tau_{\rm rec}, \tau_\alpha\}. \tag{19}$$

## Comparison with observations

The locus of possible PMF states for different values of $I_H \sim \tilde{B}^4 \lambda_B^5$ under the theory that we have described is represented by the blue-gold line in Fig. 3. We denote the largest value of $I_H$ consistent with EWPT magnetogenesis by $I_{H,\max}$; this corresponds to $\tilde{\rho}_B(t_*) = \tilde{\rho}_\gamma(t_*)$ and $\lambda_B(t_*) = r_H(t_*)$. For $I_H \lesssim 10^{-29} I_{H,\max}$, decays terminate on line (i) in Fig. 3 [Equation (40) in Methods], which represents Equation (8) with $\tau = \tau_{\rm rec}$ given by Equation (14) and $\mathrm{Pm} = \mathrm{Pm_{Sp}}$. Use of Equation (14) is valid here because $\delta_c \gtrsim r_L, d_i$ [see Equations (41) and (42) in Methods]. The Spitzer estimate of Pm is valid at recombination only if $\tilde{B} \lesssim \tilde{B}_{\rm iso} \sim 10^{-13}$ G [Equation (44) in Methods], so decays with $I_H \gtrsim 10^{-29} I_{H,\max}$ have a shorter timescale at recombination—they terminate on line (ii) [Equation (45) in Methods], which represents Equation (8) with $\tau = \tau_{\rm rec}$ given by Equation (14) and $\mathrm{Pm} \sim (r_L/\lambda_{\rm mfp})^2 \mathrm{Pm_{Sp}}$. For $I_H \gtrsim 10^{-2} I_{H,\max}$, the states on line (ii) have $\delta_c < d_i, r_L$ [see Equations (46) and (47) in Methods], so Equation (14) is invalid for them. These decays pass through line (ii) at some time before recombination with timescale given by Equation (17). However, they do access the domain of validity of Equation (14) if, before $t_{\rm recomb}$, $\tilde{B}$ becomes small enough for $\delta_c$ to be comparable with relevant kinetic scales. When that happens, their timescale becomes much larger than $t_{\rm recomb}$ so further decay is prohibited—these decays all terminate with $\tilde{B} \sim 10^{-11}$ G, which corresponds to $\delta_c \sim d_i$ at $t_{\rm recomb}$ [see Equation (46) in Methods]. Decays with $I_H \gtrsim 10^8 I_{H,\max}$ are radiation drag limited at recombination [line (iv); Equation (55) in Methods]—such decays are inconsistent with EWPT magnetogenesis, but could originate from magnetogenesis at the quantum-chromodynamic (QCD) phase transition, when $r_H \sim 10^{-6}$ Mpc[13,26].

The EGMF parameters represented by the blue-gold line are consistent with Equation (1) for $I_H \gtrsim 10^{-23} I_{H,\max}$, i.e.,

$$\left[\frac{\tilde{B}(t_*)}{10^{-5.5}\,{\rm G}}\right]^4 \left[\frac{\lambda_B(t_*)}{10^{-10}\,{\rm Mpc}}\right]^5 \gtrsim 10^{-23}. \tag{20}$$

The relic of a field with $\lambda_B(t_*) \sim 10^{-2} r_H(t_*) \sim 10^{-10}$ Mpc at the EWPT would therefore be consistent with Equation (1)—modulo any modifications for plasma instabilities in voids[17–22]—if $\tilde{\rho}_B(t_*) \gtrsim 10^{-6.5} \tilde{\rho}_\gamma(t_*)$. This confirms the assertion in the title of this paper. Intriguingly, if instead $\tilde{\rho}_B(t_*) \sim \tilde{\rho}_\gamma(t_*)$ and $\lambda_B(t_*) \gtrsim 10^{-2} r_H(t_*)$, then we find $\tilde{B} \sim 10^{-11}$ G at recombination. PMFs of this strength would provide a seed for magnetic fields in galaxy clusters that would not require significant amplification by turbulent dynamo after structure formation to reach their present day strength of $\sim\mu$G[43], although dynamo would still be required to maintain cluster fields at present levels. We emphasise that a cluster field so maintained by dynamo need not (and, in all likelihood, would not) retain memory of its primordial seed. We also note that PMFs of $10^{-11}$ G strength are considered a promising candidate to resolve the Hubble tension, by modifying the local rate of recombination[41,42].

As an aside, we note that the relevance of reconnection physics is not restricted to non-helical decay[37]. Some analogues for maximally helical PMFs of the results of this section (relevant for magnetogenesis

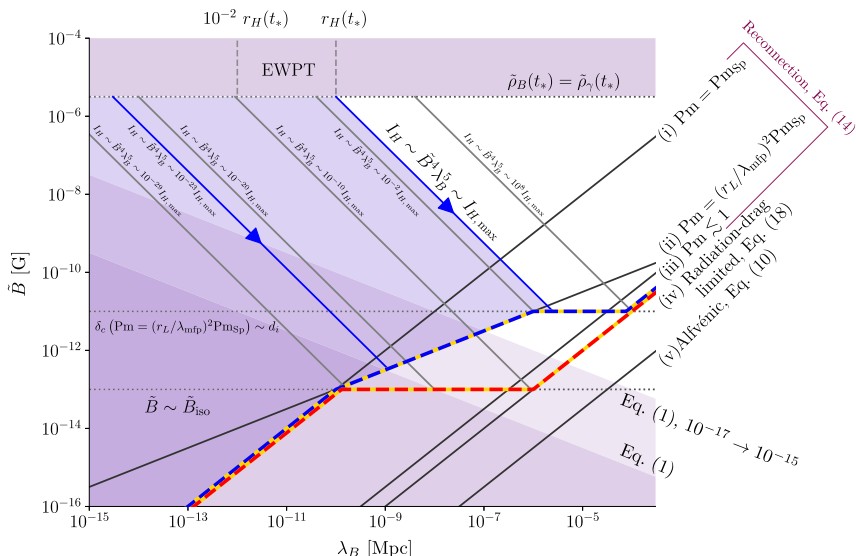

**Fig. 3 | Reconnection-controlled decay of non-helical PMFs.** As in Fig. 1, purple regions denote values of $\bar{B}$ and $\lambda_B$ excluded on physical or observational grounds [Equation (1)]. Under decays that conserve $I_H$ [Equation (11)], $\bar{B}$ and $\lambda_B$ evolve along lines parallel to the ones shown in blue. The predicted values of modern-day $\bar{B}$ and $\lambda_B$ are given by the intersection of these lines with Equation (8) evaluated at recombination [represented by lines (i–v), which are derived in Methods], with $\tau$ the prevailing decay timescale. The blue-gold line shows the locus of possible present-day states resulting from reconnection-controlled decays on the time-scales explained in the main text, assuming that the microscopic viscosity of the primordial plasma was controlled by collisions between protons. The effective value of Pm in Equation (14) might have been heavily suppressed when $\bar{B} > \bar{B}_{iso}$ if viscosity were then instead governed by plasma microinstabilities—the red-gold line shows the locus of modern-day states corresponding to the extreme choice of Pm $\lesssim 1$ for $\bar{B} > \bar{B}_{iso}$. In either case, we see that PMFs generated at the EWPT with a wide range of values of $I_H$ produce modern-day relics that are consistent with Equation (1), and even with the stronger version of this constraint [see text below Equation (1)] which is indicated by the pale purple region.

mechanisms capable of parity violation) are presented in Section A of the Supplementary Information.

### Role of plasma microinstabilities
Finally, we note that, for $\bar{B} > \bar{B}_{iso}$, the effective values of $\tilde{\nu}$ and $\tilde{\eta}$ might be dictated by plasma microinstabilities rather than by collisions between protons[65] (this is conjectured to happen in galaxy clusters[66]). In Methods, we show that the decay of the integral-scale magnetic energy is too slow to excite the firehose instability that is important in the cluster context [see Equation (60)]. Nonetheless, we cannot rule out other microinstabilities—for example, the excitation of the mirror instability by reconnection has been studied by ref. 67, although its effect on the rate of reconnection remains unclear. The most dramatic effect that microinstabilities in general could plausibly have would be to reduce the effective value of Pm to $\lesssim 1$ if $\bar{B} > \bar{B}_{iso}$ (see refs. 68,69). This corresponds to the red-gold line in Fig. 3, which remains consistent with Equation (1) for $I_H \gtrsim 10^{-20} I_{H,max}$. Compatibility between the EWPT-magnetogenesis scenario and the observational constraints on EGMFs therefore appears robust.

## Methods
### Post-recombination evolution
In the matter-dominated Universe after recombination, the transformation that maps Minkowski spacetime MHD onto its expanding Universe equivalent is not Equation (3), but[24]

$$\tilde{\rho} = a^3 \rho, \qquad \tilde{p} = a^4 p, \quad \tilde{\mathbf{B}} = a^2 \mathbf{B}, \quad \tilde{\mathbf{u}} = a^{1/2} \mathbf{u},$$
$$\tilde{\eta} = \eta/a^{1/2}, \quad \tilde{\nu} = \nu/a^{1/2}, \quad d\tilde{t} = dt/a^{1/2}. \qquad (21)$$

As $a \propto t^2$ in the matter-dominated Universe, $\tilde{t} \propto \log t$, so a power-law decay in rescaled variables corresponds to only a logarithmic decay in comoving variables[14]. Thus, in computing the expected present-day strength of EGMFs, one may assume the decay of $\bar{B}$ to terminate at recombination with negligible error.

### Derivation of Equation 10
In order to apply Equation (8), we require an expression for the conformal time at recombination, $t_{recomb}$. From the Friedmann equation,

$$\frac{1}{a^4} \left( \frac{da}{dt} \right)^2 = \frac{8\pi G \rho}{3}, \qquad (22)$$

where $G$ is the gravitational constant, the entropy equation

$$g T^3 a^3 = \text{const}, \qquad (23)$$

where $g$ is the number of degrees of freedom of the radiation field and $T$ is the temperature, and Stefan's law for the radiation density

$$\rho = 3\chi g T^4, \qquad (24)$$

where $\chi = \pi^2/90 c^5 \hbar^3$ (we work in energy units for temperature, with Boltzmann constant $k_B = 1$), it can be shown that

$$\left( \frac{dT}{dt} \right)^2 = 8\pi G g_0 \chi T^4 T_0^2 \left( \frac{g}{g_0} \right)^{1/3}, \qquad (25)$$

where the subscript 0 refers to quantities evaluated at the present day. Because $(g/g_0)^{1/6} \simeq 1$, one may solve Equation (25) to give an expression for the cosmic temperature as a function of conformal time,

$$T = \frac{1}{t T_0} \sqrt{\frac{1}{8\pi G g_0 \chi}}. \qquad (26)$$

With $g_0 = 2$ (for the two photon-polarisation states), one obtains

$$t \sim 10^{16.5}\,\text{s}\left(\frac{T}{0.3\text{eV}}\right)^{-1}. \tag{27}$$

Therefore, Equation (8) becomes

$$\tau \sim 10^{16.5}\,\text{s}\left(\frac{T}{0.3\text{eV}}\right)^{-1}. \tag{28}$$

Thus, $t_{\text{recomb}} \sim 10^{16.5}$ s. Equation (28) can be used to relate $\tilde{B}$ and $\lambda_B$ under the assumption that the decay occurs on the Alfvénic timescale $\tau \sim \lambda_B/\tilde{v}_A$ [Equation (9)]. As noted in the main text, $\tilde{v}_A$ should be computed using the baryon density $\tilde{\rho}_b$, because the photon mean free path[13]

$$\lambda_{\text{mfp},\gamma} = \frac{1}{a\sigma_T n_e} \sim 1\text{Mpc}\left(\frac{T}{0.3\text{eV}}\right)^{-2} \tag{29}$$

(where $\sigma_T$ is the Thompson-scattering cross-section) is large compared with $\lambda_B$ at the time of recombination, indicating that photons are not strongly coupled to the fluid[45]. However, because $\tilde{\rho}_b \simeq \tilde{\rho}_\gamma$ at the time of recombination, the decoupling of photons does not affect Equation (10). The Alfvén speed is

$$\tilde{v}_A = \frac{\tilde{B}}{\sqrt{4\pi\tilde{\rho}_b}} \simeq 10^{16}\,\text{cm s}^{-1}\frac{\tilde{B}}{1G}\left(\frac{T}{0.3\text{MeV}}\right)^{1/2}, \tag{30}$$

where we have used $\tilde{\rho}_b = a^4\rho_b \simeq a^4 m_i n_b$, with $m_i$ the proton mass and $n_b$ the WMAP value for the baryon number density $n_b \simeq 2.5\times10^{-7}\,\text{cm}^{-3}a^{-3}$[70], and taken $a \simeq T_0/T$ [Equation (23)]. Comparing Equation (9) and Equation (28), and substituting Equation (30), we have

$$\tilde{B} \sim 10^{-8.5}\,\text{G}\left(\frac{\lambda_B}{1\,\text{Mpc}}\right)\left(\frac{T}{0.3\text{eV}}\right)^{1/2}. \tag{31}$$

Evaluated at $T = T(t_{\text{recomb}}) = 0.3$ eV, this is Equation (10).

**Derivation of line (i) of Fig. 3**
Line (i) represents Equation (14) evaluated at the time of recombination $t_{\text{recomb}}$, with $\text{Pm} = \text{Pm}_{\text{Sp}} \equiv \tilde{v}_{\text{Sp}}/\tilde{\eta}_{\text{Sp}}$, where $\tilde{v}_{\text{Sp}}$ and $\tilde{\eta}_{\text{Sp}}$ are the comoving Spitzer values of kinematic viscosity and magnetic diffusivity respectively[59]. We first evaluate $\text{Pm}_{\text{Sp}}$.

Under Spitzer theory, the dominant component of the plasma viscosity at the scale of the rate-determining current sheet is due to ion-ion (i.e., proton–proton) collisions. The collision frequency is[59]

$$\nu_{ii} \sim \frac{e^4 n_i \ln\Lambda_{ii}}{m_i^{1/2} T_i^{3/2}}, \tag{32}$$

where $e$ is the elementary charge, $n_i$ the ion number density, $m_i$ the ion mass, $T_i$ the ion temperature, and $\ln\Lambda_{ii}$ the Coulomb logarithm for ion-ion collisions. Neglecting any anisotropising effect of the magnetic field (see main text), the comoving isotropic kinematic viscosity is[71]

$$\tilde{v}_{\text{Sp}} \sim \frac{v_{\text{th},i}^2}{a\nu_{ii}} \sim \frac{T_i^{5/2}}{am_i^{1/2}e^4 n_i \ln\Lambda_{ii}} \sim 10^{18}\,\text{cm}^2\text{s}^{-1}\left(\frac{T}{0.3\text{eV}}\right)^{1/2}, \tag{33}$$

where $v_{\text{th},i} = \sqrt{2T_i/m_i}$ is the thermal speed of ions, and we have assumed $T_i \simeq T$, used $a \simeq T_0/T$ [Equation (23)], taken $n_i$ to be equal to the WMAP value for the baryon number density $n_b \simeq 2.5\times10^{-7}\,\text{cm}^{-3}a^{-3}$[70], and estimated the Coulomb logarithm $\ln\Lambda_{ii}$ by

$$\ln\Lambda_{ii} \simeq \ln\frac{T_i^{3/2}}{e^3 n_i^{1/2}} \simeq 20. \tag{34}$$

Similarly, the electron-ion collision frequency is[71]

$$\nu_{ei} \sim \frac{e^4 n_e \ln\Lambda_{ei}}{m_e^{1/2} T_e^{3/2}}, \tag{35}$$

where $n_e \simeq n_i$ is the electron number density, $T_e$ the electron temperature, and $\ln\Lambda_{ei}$ the Coulomb logarithm for electron-ion collisions. Equation (35) leads to the Spitzer[59] value for the magnetic diffusivity

$$\tilde{\eta}_{\text{Sp}} \sim \frac{\nu_{ei}m_e c^2}{4\pi n_e e^2 a} \sim 10^{10.5}\,\text{cm}^2\text{s}^{-1}\left(\frac{T}{0.3\text{eV}}\right)^{-1/2}, \tag{36}$$

where we have used $\ln\Lambda_{ei} \simeq \ln\Lambda_{ii} \simeq 20$, assumed the electron temperature $T_e \simeq T$, and again neglected any anisotropy resulting from the magnetic field. From Equations (33) and (36), we have

$$\text{Pm}_{\text{Sp}} = \frac{\tilde{v}_{\text{Sp}}}{\tilde{\eta}_{\text{Sp}}} \sim \frac{T^4}{m_e^{1/2}m_i^{1/2}e^6 n_i \ln\Lambda_{ii}\ln\Lambda_{ei}} \sim 10^7\left(\frac{T}{0.3\,\text{eV}}\right). \tag{37}$$

Let us now evaluate the Lundquist number, Equation (15), in order to compare it with $S_c$, as Equation (14) requires. Note that, as above, it is the Alfvén speed based on baryon inertia that appears in Equation (15); photons are even more weakly coupled to the cosmic fluid at reconnection scales than at scale $\lambda_B$ as the former are typically small compared with the latter. Using Equations (13), (30), and (37), we find the Lundquist number

$$\begin{aligned}S = {}& \frac{1}{\sqrt{1+\text{Pm}_{\text{Sp}}}}\frac{\tilde{v}_A(t_*)\lambda_B(t_*)}{\tilde{\eta}}\left[\frac{\lambda_B(t_*)}{\lambda_B}\right]^{1/4} \sim 10^9\left[\frac{\tilde{B}(t_*)}{10^{-5.5}\,\text{G}}\right]\left[\frac{\lambda_B(t_*)}{10^{-12}\,\text{Mpc}}\right]\\ & \times\left[\frac{T}{0.3\text{eV}}\right]^{1/2}\left[\frac{\lambda_B(t_*)}{\lambda_B}\right]^{1/4}.\end{aligned} \tag{38}$$

Equation (38) shows that $S \gg S_c \sim 10^4$ [unless $\tilde{B}(t_*)$ or $\lambda_B(t_*)$ are very small, in which case their evolution is inconsistent with the observational constraint (1), so we neglect this possibility for simplicity]. Substituting Equation (37), we find that the decay timescale (14) is

$$\tau \sim 10^{5.5}\left(\frac{T}{0.3\text{eV}}\right)^{1/2}\frac{\lambda_B}{\tilde{v}_A}. \tag{39}$$

Comparing Equations (28) and (39), and again substituting Equation (30), we find

$$\tilde{B} \sim 10^{-3}\,\text{G}\left(\frac{\lambda_B}{1\,\text{Mpc}}\right)\left(\frac{T}{0.3\,\text{eV}}\right). \tag{40}$$

Evaluated at $T = T(t_{\text{recomb}}) = 0.3$ eV, this is line (i) of Fig. 3.

Finally, we note that when reconnection occurs under large-Pm conditions with isotropic Spitzer viscosity, the ratio of $\delta_c$ [Equation (16)] to $r_L$ [defined below Equation (15)] prior to recombination is independent of the magnetic-field strength, temperature and density:

$$\frac{\delta_c}{r_L} \sim S_c^{1/2}\left(\frac{m_e}{m_i}\right)^{1/4} \sim 10, \tag{41}$$

where we have used Eqs (36), (37) and (30). Thus, $\delta_c > r_L$ always. Furthermore, we find from Equations (15), (16), (30), (36), (37) and the definition of $d_i$ [see below Equation (16)] that

$$\frac{\delta_c}{d_i} \sim S_c^{1/2}\left(\frac{m_e}{m_i}\right)^{1/4}\frac{v_{\text{th},i}}{\tilde{v}_A} \sim \left(\frac{\tilde{B}}{10^{-9}\,\text{G}}\right)^{-1}. \tag{42}$$

Therefore, $\delta_c > d_i, r_L$ at recombination for all relevant field strengths, so we are justified in using fluid theory to describe decays with $\tilde{B} < \tilde{B}_{\text{iso}}$ [evaluated in Equation (44)].

As described in the main text, Equation (40) is valid when $\tilde{B}$ is small enough for the Larmor radius of ions $r_L$ to be larger than their mean free path

$$\lambda_{\text{mfp}} \sim \frac{v_{\text{th},i}}{\nu_{ii} a} \sim 10^{12}\,\text{cm}. \tag{43}$$

The critical magnetic field strength above which this condition is no longer satisfied is

$$\tilde{B}_{\text{iso}} \sim \frac{m_i c \nu_{ii} a^2}{e} \sim 10^{-13}\,\text{G} \left(\frac{T}{0.3\,\text{eV}}\right)^{-1/2}. \tag{44}$$

### Derivation of line (ii) of Fig. 3

Line (ii) represents Equation (14) evaluated at the time of recombination $t_{\text{recomb}}$, with magnetic Prandtl number $\text{Pm} \sim (r_L/\lambda_{\text{mfp}})^2 \text{Pm}_{\text{Sp}} = (\tilde{B}_{\text{iso}}/\tilde{B})^2 \text{Pm}_{\text{Sp}}$. Note that this suppression of Pm relative to $\text{Pm}_{\text{Sp}}$ *increases* the value of $S$ at any given $\bar{v}_A$ and $\lambda_B$ relative to the value (38) of $S$ that corresponds to $\text{Pm} = \text{Pm}_{\text{Sp}}$. We therefore expect this family of decays also to have $S \gg S_c \sim 10^4$.

The inclusion of the factor of $(\tilde{B}_{\text{iso}}/\tilde{B})^2$ in Pm modifies Equation (40) straightforwardly: it becomes

$$\tilde{B} \sim 10^{-3}\,\text{G} \left(\frac{\tilde{B}_{\text{iso}}}{\tilde{B}}\right)\left(\frac{\lambda_B}{1\,\text{Mpc}}\right)\left(\frac{T}{0.3\,\text{eV}}\right).$$
$$\Rightarrow \tilde{B} \sim 10^{-8}\,\text{G} \left(\frac{\lambda_B}{1\text{Mpc}}\right)^{1/2}\left(\frac{T}{0.3\,\text{eV}}\right)^{1/4}. \tag{45}$$

Evaluated at $T = T(t_{\text{recomb}}) = 0.3$ eV, this is line (iv) of Fig. 3.

The analogue of Equation (42) for $\text{Pm} \sim (\tilde{B}_{\text{iso}}/\tilde{B})^2 \text{Pm}_{\text{Sp}}$ is

$$\frac{\delta_c}{d_i} \sim S_c^{1/2}\left(\frac{m_e}{m_i}\right)^{1/4}\frac{v_{\text{th},i}}{\bar{v}_A}\frac{\tilde{B}_{\text{iso}}}{\tilde{B}} \sim \left\{\tilde{B}\Big/\left[10^{-11}\,\text{G}\left(\frac{T}{0.3\,\text{eV}}\right)^{-1/4}\right]\right\}^{-2}, \tag{46}$$

while the corresponding analogue of Equation (41) is

$$\frac{\delta_c}{r_L} \sim S_c^{1/2}\left(\frac{m_e}{m_i}\right)^{1/4}\frac{\tilde{B}_{\text{iso}}}{\tilde{B}} \sim \left\{\tilde{B}\Big/\left[10^{-12}\,\text{G}\left(\frac{T}{0.3\,\text{eV}}\right)^{-1/2}\right]\right\}^{-1}. \tag{47}$$

Equation (46) shows that $\delta_c \gtrsim d_i$ at $t_{\text{recomb}}$ if $\tilde{B} \lesssim 10^{-11}$G, while Equation (47) indicates that $\delta_c \gtrsim r_L$ if $\tilde{B} \lesssim 10^{-12}$G. Following the prescription described in[55], we use the former condition on $\tilde{B}$ as the domain of validity of Equation (14) in Fig. 3, though we note that our results do not depend strongly on this choice—the order-of-magnitude difference between the two critical values of $\tilde{B}$ is comparable to the degree of accuracy to which our scaling arguments are valid.

We also note that the temperature dependence of Equation (46) means that a decaying field that developed $\delta_c \gtrsim d_i$ *before* recombination would have done so at a field strength $\tilde{B} < 10^{-11}$G; strictly, therefore, the decay of primordial fields should terminate somewhere below the horizontal part of the blue-gold curve in Fig. 3, not directly on it. However, the difference is order unity and thus negligible for the purposes of our order-of-magnitude estimates. This is because magnetic decay was strongly suppressed by radiative drag at early times [a consequence of the strong temperature dependence of Equation (55)] —i.e., when temperatures exceeded around $10^2 \times 0.3$ eV. For all relevant values of $l_H$, the magnetic-field strength would therefore have greatly exceeded the critical value required for $\delta_c \sim d_i$ until the time that corresponds to this temperature, and by that time the critical field strength indicated by Equation (46) was already within a small factor of its value at recombination.

### Derivation of line (iii) of Fig. 3

Line (iii) represents Equation (14) at the time of recombination $t_{\text{recomb}}$, with $\text{Pm} \lesssim 1$. With $\text{Pm} \lesssim 1$, Equation (38) should be replaced by

$$S \sim 10^{12.5} \left[\frac{\tilde{B}(t_*)}{10^{-5.5}\,\text{G}}\right]\left[\frac{\lambda_B(t_*)}{10^{-12}\,\text{Mpc}}\right] \times \left[\frac{T}{0.3\,\text{eV}}\right]\left[\frac{\lambda_B(t_*)}{\lambda_B}\right]^{1/4}, \tag{48}$$

so that $S \gg S_c \sim 10^4$ for all decays of interest. The decay timescale (14) therefore becomes

$$\tau \simeq 10^2 \frac{\lambda_B}{\bar{v}_A}. \tag{49}$$

Comparing Equations (28) and (39), and substituting Equation (30), we find

$$\tilde{B} \sim 10^{-6.5}\,\text{G} \left(\frac{\lambda_B}{1\,\text{Mpc}}\right)\left(\frac{T}{0.3\,\text{eV}}\right)^{1/2}. \tag{50}$$

Evaluated at $T = T(t_{\text{recomb}}) = 0.3$ eV, this is line (iii) of Fig. 3.

The analogues of Equations (42) and (41) for $\text{Pm} \lesssim 1$ (but $\tilde{\eta} \sim \tilde{\eta}_{\text{Sp}}$) are

$$\frac{\delta_c}{r_L} \sim S_c^{1/2}\frac{c}{v_{\text{th},e}}\frac{\ln \Lambda_{ei}}{\Lambda_{ii}} \sim 10^{-2.5}\left(\frac{T}{0.3\,\text{eV}}\right)^{-1/2}, \tag{51}$$

and

$$\frac{\delta_c}{d_i} \sim S_c^{1/2}\frac{c}{\bar{v}_A}\left(\frac{m_e}{m_i}\right)^{1/2}\frac{\ln \Lambda_{ei}}{\Lambda_{ii}} \sim \left\{\tilde{B}\Big/\left[10^{-13}\,\text{G}\left(\frac{T}{0.3\,\text{eV}}\right)^{-1/2}\right]\right\}^{-1}. \tag{52}$$

Note that the field strength at which $\delta_c \sim d_i$ is approximately equal to $\tilde{B}_{\text{iso}}$ at recombination (both are $\sim 10^{-13}$ G), while $\delta_c \ll r_L$. The red-gold line in Fig. 3 therefore extends past line (iii) to line (iv) along the line $\tilde{B} \sim \tilde{B}_{\text{iso}}$.

### Radiation drag and the derivation of line (iv) of Fig. 3

As well as by viscosity arising from collisions between ions, the kinetic energy of primordial plasma flows (after neutrino decoupling) can be dissipated by electron–photon collisions (Thompson scattering). Around the time of recombination, the comoving mean free path of photons, Equation (29), is much larger than the anticipated correlation scale of the magnetic field (and, therefore, of any magnetically driven flows). Under these conditions, the effect of Thompson scattering is to induce a drag on electrons. Owing to the collisional coupling between ions and electrons, this drag can dissipate bulk plasma flows.

The comoving drag force on the fluid per unit baryon density is

$$\bar{\mathbf{F}}_D = -\tilde{\alpha}\tilde{\mathbf{u}}, \tag{53}$$

where[24]

$$\tilde{\alpha} \sim \frac{c}{\lambda_{\text{mfp},\gamma}}\frac{\rho_\gamma}{\rho_b} \sim 10^{-13.5}\,\text{s}^{-1}\left(\frac{T}{0.3\,\text{eV}}\right)^3. \tag{54}$$

As explained in the main text, the effect of drag is most important at the scale $\lambda_B$ (it becomes increasingly subdominant to magnetic tension at smaller scales) where it inhibits inflows to the reconnection layer. When the timescale $\tau_\alpha \equiv \tilde{\alpha}\lambda_B^2/\bar{v}_A^2$ on which flux can be delivered to the layer by strongly dragged inflows is larger than the reconnection timescale of the critical sheet $\tau_{\text{rec}}$ [see Equation (19)], $\tau_\alpha$ gives the

timescale for energy decay. Equation (28) with $\tau = \tau_\alpha$ yields, after substitution of Equations (30) and Equation (54)

$$\tilde{B} \sim 10^{-7} \, \mathrm{G} \left( \frac{\lambda_B}{1\,\mathrm{Mpc}} \right) \left( \frac{T}{0.3\,\mathrm{eV}} \right)^{3/2}. \tag{55}$$

Evaluated at $T = T(t_{\mathrm{recomb}}) = 0.3$ eV, this is line (iv) of Fig. 3.

### Non-excitation of the firehose instability

Plasma with an anisotropic viscosity tensor can, in principle, be unstable to a variety of instabilities that develop at kinetic scales. For a decaying magnetic field, an instability of particular importance is the firehose, which can generate the growth of small-scale magnetic fields in response to the decay of large-scale ones[65,72]. This happens if the size of the (negative) pressure anisotropy $\Delta$ exceeds a critical value:

$$\Delta \equiv \frac{p_\perp - p_\parallel}{p_\parallel} \le -\frac{2}{\beta_i} \tag{56}$$

where $p_\parallel$ and $p_\perp$ are the thermal pressures parallel and perpendicular to the magnetic field, and

$$\beta_i \equiv \frac{p_\parallel}{B^2 / 8\pi} \tag{57}$$

is the plasma beta. $\Delta$ can be estimated as[65]

$$\Delta \sim \frac{1}{\nu_{ii}} \frac{1}{B} \frac{dB}{d\bar{t}} \sim -\frac{1}{a \nu_{ii} \tau} \sim -10^{-11} \left( \frac{T}{0.3\,\mathrm{eV}} \right)^{1/2}, \tag{58}$$

where $\bar{t}$ is cosmic time [defined below Equation (2)]. Naturally, the value of $\beta_i$ at any given $T$ depends on the evolution of the magnetic field. A lower bound on the value of $\tilde{B}$ at any given time for a given initial condition is the one that would develop from a decay on the kinetic reconnection timescale, $\tau \sim 10 \lambda_B / \bar{v}_A$ [Equation (17)]. Solving Equations (13), (17), (28), and (30) simultaneously, we find that this is

$$\tilde{B}(t) \sim 10^{-13} \, \mathrm{G} \left( \frac{T}{0.3\,\mathrm{eV}} \right)^{5/18} \times \left[ \frac{\lambda_B(t_*)}{10^{-12}\,\mathrm{Mpc}} \right]^{5/9} \left[ \frac{\tilde{B}(t_*)}{10^{-5.5}\,\mathrm{G}} \right]^{4/9}. \tag{59}$$

Using this lower bound on $\tilde{B}$, we can obtain an upper limit on $|\beta_i \Delta|$:

$$|\beta_i \Delta| \lesssim 10^{-6} \left( \frac{T}{0.3\,\mathrm{eV}} \right)^{-1/18} \times \left[ \frac{\lambda_B(t_*)}{10^{-12}\,\mathrm{Mpc}} \right]^{-10/9} \left[ \frac{\tilde{B}(t_*)}{10^{-5.5}\,\mathrm{G}} \right]^{-8/9}. \tag{60}$$

Equation (60) suggests that the threshold for instability (56) is never met, unless $\lambda_B(t_*)$ and/or $\tilde{B}(t_*)$ are so small as to be inconsistent with the observational constraint (1).

### Numerical simulation

The numerical simulations visualised in Fig. 2 and described in Section B of the Supplementary Information were conducted using the spectral MHD code Snoopy[73]. The code solves the equations of incompressible MHD in Minkowski spacetime with hyper-viscosity and hyper-resistivity both of order $n$, viz.,

$$\frac{\partial \mathbf{u}}{\partial t} + \mathbf{u} \boldsymbol{\nabla} \mathbf{u} = -\boldsymbol{\nabla} p + (\boldsymbol{\nabla} \times \mathbf{B}) \times \mathbf{B} - (-1)^{n/2} \nu_n \nabla^n \mathbf{u},$$
$$\frac{\partial \mathbf{B}}{\partial t} = \boldsymbol{\nabla} \times (\mathbf{u} \times \mathbf{B}) - (-1)^{n/2} \eta_n \nabla^n \mathbf{B}, \tag{61}$$

where $p$, the thermal pressure, is determined via the incompressibility condition

$$\boldsymbol{\nabla} \mathbf{u} = 0. \tag{62}$$

Snoopy uses a pseudo-spectral algorithm with a 2/3 dealiasing rule. It performs time integration of non-dissipative terms using a low-storage, third-order, Runge-Kutta scheme, while solving the dissipative terms using an implicit method that preserves the overall third-order accuracy of the numerical scheme. In all runs presented here, we employ $\nu_n = \eta_n = 10^{-12}$, $n = 6$ and use a resolution of $512^3$. The size of the periodic simulation domain is $L_{\mathrm{box}} = 2\pi$.

## Data availability

Source data for Fig. 2 and Supplementary Fig. 2 are provided in this paper. The datasets generated during and/or analyzed during the current study are available from the corresponding author upon request. Source data are provided with this paper.

## Code availability

The simulations presented in this paper were conducted using the publicly available code Snoopy[73]. The codes that were used to generate the figures in the paper are available from the corresponding author upon request.

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

## Acknowledgements

We are grateful to R.B., B.C., F.R., and D.U. for stimulating discussions, and to K.S. for posing a question that led us to write Section B of the Supplementary Information. D.N.H. was supported by a UK STFC studentship. The work of A.A.S. was supported in part by the UK EPSRC grant EP/R034737/1. This work used the ARCHER UK National Supercomputing Service (http://www.archer.ac.uk).

## Author contributions

D.N.H. conducted the study and wrote the manuscript, and A.A.S. provided conceptual advice and comments on the manuscript.

## Competing interests

The authors declare no competing interests.
