## [Peer Review File · Nature Communications]

Cosmic-void observations reconciled with primordial magnetogenesisEditorial Note: Parts of this peer review file have been redacted as indicated to maintain the confidentiality of other journals.

REVIEWER COMMENTS

Reviewer #1 (Remarks to the Author):

The revised version of the manuscript takes into account most of my comments from the first review cycle and the authors have answered questions that I've asked in my first report.

I am still puzzled by the status of Eq. (17), as a "robust numerical result whose theoretical explanation is an active research topic". I am not sure if the numerical modelling mentioned here was ever done for conditions valid in the early universe and on time scales relevant to the cosmological magnetic field..... But I understand this limitation and I think it's Ok to leave it like this. Maybe, to illustrate the uncertainty of the status of (17), the authors might consider to add alternative locus of endpoints of cosmological evolution at "Alfvénic" line in Fig. 3.

Otherwise, I think the manuscript can be published in its present form.

Reviewer #2 (Remarks to the Author):

I have read the paper with interest and have also looked through the communication with earlier referees. It utilizes recent results in MHD theory and combines them into a comprehensive assessment of the magnetic field levels that can be expected in the voids at the present time. As I have stated below, it is more of an assembly of recent results and applying them to cosmology rather than new per se.

I think the paper is basically publishable, but there are several points where the paper seems to claim more than is justified. Below my detailed comments in the order they appear.

1. In the abstract, I think it would be important to clarify when the authors are talking about nonhelical or helical fields.

2. Regarding the lower limits, as already emphasized by an earlier referee, it would be important to spell out that those limit are subject to the assumption that plasma instabilities are not important. The recent review by Addazi+22 has some recent assessment on this. Since this is potentially a major threat, I suggest a mention this in several critical places in the paper.

3. The discussion of the magnetic version of the Loitsyansky integral under "Results" on p. 2-3 seems excessive and was never seen in MHD simulations. As the authors discuss only later, simulations have shown a slower decay, and the point of HS21 is that it clarifies the reason behind it. I think this should be reflected better in a revised presentation. Since the work of HS21 is fairly recent, any additional confirmation of their results are of course welcome.

4. Eq. (4) is only correct when $n=2, 6$, etc. The authors may want to add a $-(-1)^{\{n/2\}}$ factor.

Reviewer #3 (Remarks to the Author):

I am afraid that I cannot recommend the paper for publications,

First, I do not see enough justification to transfer the manuscript from [redacted] to Nature Communications. I do not see the novelty of the manuscript as well as importance for a wide audience to be published in Nature.

Second, I disagree with authors that they replied to my previous comments. Especially when addressing the definition of the Alfvén speed: I suggest to re-address carefully the paper by Hu & White, 1997, <https://journals.aps.org/prd/abstract/10.1103/PhysRevD.56.596>

in order to express coupling between baryons and photons. Importantly, the Alfvén speed in such a definition will be time dependent ($B^2 \propto 1/a^4$ and $\rho_b \propto 1/a^3$), see their Eq. 30 – and correspondingly it must be properly accounted when applying the comoving quantities for numerical simulations.

Third, I would like to highlight the magnetic field total energy density can be several order of magnitudes larger than it was estimated previously (and without the formalism described by the Authors) if we just account the decay laws and proper consideration of the Big Bang nucleosynthesis limits,

See Fig. 1 of <https://journals.aps.org/prl/abstract/10.1103/PhysRevLett.128.221301>

I have some concerns about other parts of the manuscript:

Eq. 1 – λ is the energy containing length scale, not a correlation length. In fact, it is assumed that the magnetic field is described as an homogeneous characterized by a λ -scale (i.e. monochromatic) field, and B_λ vs λ physically differs from the magnetic field spectral description. I also believe that the causality requirements for the scale-invariant magnetic field works properly (the field is "frozen-in" at super-horizon scales while the field is decaying in sub-horizon scales).

Additional Comments

I do not think that the Authors replied to Comments by Referee #2:

Comment:

I have now gone through the paper "Cosmic-void observations reconciled with primordial magnetogenesis" by Hosking and Schekhochihin. The authors combine their theory for inverse

transfer in non-helical decaying magnetically dominated turbulence and a notion from previous authors (and themselves) that the time-scale for this could be the reconnection time. They argue that this allows one to explain magnetic fields in voids as arising from electroweak phase transitions. All the pieces of this argument are at present fairly speculative. There is considerable controversy whether the gamma-ray observations do indicate magnetic fields in the voids or can rather be explained as due to plasma instabilities (Broderick, Chang, Pfrommer, 2012, ApJ, 752, 22) and the large body of subsequent literature. The strength and correlation scale of magnetic fields generated during the EWPT is uncertain, even whether the phase transition can be first order (the reviews cited by the authors), and the theory of Hosking and Schekhochihin is still being scrutinized critically by the community. Moreover, even without invoking reconnection time scale as being the relevant one for the decay, there exist work which suggests that if all the above conditions are valid, the void magnetic fields can arise from the EWPT (Brandenburg et al, 2017, Phys. Rev. D 96, 123528), which does not agree with the work of Wagstaff and Banerjee, precisely because non helical decay was found to be slower than assumed by the later authors. Thus I find the current paper somewhat speculative, and also not novel enough nor of such immediate interest to a large astronomical community. Thus I am not able to recommend it for publication in [redacted]. ”

The Authors Answer:

We thank the referee for their consideration of our manuscript and their comments on it. We have now acknowledged the alternative view of Broderick et. al. (2012) in our manuscript. We note that our work differs from Brandenburg et. al. 2017 (which we have also now cited) in that (i) we provide a theoretical explanation for the inverse transfer via the Saffman helicity invariant, and (ii) include the role of the magnetic reconnection timescale. As such, our quantitative predictions are different from those of the earlier work — our theory allows EWPT fields generated with scale significantly smaller than the Hubble radius, for example.

The presence of the magnetic fields is still under debates: the Authors did not discuss in details arguments for the large-scale magnetic fields in voids; For example they do refer to Briderick et al (2012) but discard morrecent studies,

- E. Broderick, et al. “Missing Gamma-ray Halos and the Need for New Physics in the Gamma-ray Sky,” *Astrophys. J.*, 868, 87, 2018.
- R. Alves Batista, et al. “The Impact of Plasma Instabilities on the Spectra of TeV Blazars,” *Mon. Not. Roy. Astron. Soc.*, 489, 3836, 2019.
- R. Perry and Y. Lyubarsky, “The role of resonant plasma instabilities in the evolution of blazar induced pair beams,” *Mon. Not. Roy. Astron. Soc.*, vol. 503, 2215, 2021

In fact they do not compare the cosmological magnetic field approach vs. plasma instability in the context of the filling factor, see Ref. Dolag et. al. "Lower Limit on the Strength and Filling Factor of Extragalactic Magnetic Fields", *Astrophys. J. Lett.* 727, L4 (2011). In addition, there is no discussion concerning the time-delay effect, see Ref. M. Ackermann et al., "The Search for Spatial Extension in High-latitude Sources Detected by the Fermi Large Area Telescope," *Astrophys. J. Suppl.*, 237, 32, 2018

It is unclear physical interpretation of λ_B in their Eq. 1. Indeed, the lower limit of the magnetic field in Eq 1 is the effective magnetic field strength that corresponds to the homogeneous field strength that is mathematically determined by the monochromatic spectrum, i.e. in the momentum space the spectrum can be approximated by $\delta(k-k_B)$ with $k_B \sim \lambda_B^{-1}$ so it is not correct to identify as the correlation length as λ_B .

The Saffman helicity invariant has been studied and discussed earlier in the literature, including works by the Authors, see Ref. D. N. Hosking and A. A. Schekochihin, "Reconnection Controlled Decay of Magnetohydrodynamic Turbulence and the Role of Invariants", *Phys. Rev. X* 11, 041005 (2021).

Comment:

It is also surprising to say that these fields can explain cluster magnetism without dynamos, given the large body of work that shows dynamos to be essential to maintain cluster magnetic fields; perhaps the authors mean that these fields would provide a seed field?

The Authors Answer:

The referee makes a good point that the phrasing of our comment was not clear. Our intended meaning was that, while dynamos are indeed essential for maintaining cluster magnetic fields, they may not be necessary for growth, as the "seed" field would already be strong enough (after compression during structure formation) to explain the observations. We have revised the manuscript to make this meaning clear.

The statement by the Authors is misleading: at large scales (voids) the magnetic fields are unchanged ("frozen-in") during the structure formation, while in the galaxy scales the observed magnetic fields are determined by non-linear processes including dynamos. The strong enough seed magnetic fields might be ruled the Faraday rotation observations, see Ref. R. Banerjee and K. Jedamzik, "Evolution of

Cosmic Magnetic Fields: From the Very Early Universe, to Recombination, to the Present”, Phys. Rev. D 70, 123003 (2004).

I also think they did not address Referee #2 other comments.

Cosmic-void observations reconciled with primordial magnetogenesis – replies to referees

D. N. Hosking, A. A. Schekochihin

April 28, 2023

We thank the referees for providing further comments on our manuscript. Our responses are below. The changes that we have made to our manuscript in response to these comments are highlighted in blue in the resubmitted article files.

Referee 1

“The revised version of the manuscript takes into account most of my comments from the first review cycle and the authors have answered questions that I’ve asked in my first report.”

“I am still puzzled by the status of Eq. (17), as a “robust numerical result whose theoretical explanation is an active research topic”. I am not sure if the numerical modelling mentioned here was ever done for conditions valid in the early universe and on time scales relevant to the cosmological magnetic field..... But I understand this limitation and I think it’s Ok to leave it like this. May be, to illustrate the uncertainty of the status of (17), the authors might consider to add alternative locus of endpoints of cosmological evolution at “Alfvénic” line in Fig. 3.”

“Otherwise, I think the manuscript can be published in its present form.”

We are grateful to the referee for their comments. We are glad that they consider the manuscript suitable for publication.

While we agree that one could wonder whether the numerically derived scaling (17) is guaranteed to hold under conditions relevant to early universe, the factor of 10 that appears in (17) but not in the Alfvénic scaling (10) is not critical for the conclusions of this article. This is because, whenever microphysical processes would tend to produce “fast” reconnection in the sense of (10) or (17), the reconnection rate would be constrained by the radiation drag on inflows (18) instead. The locus of endpoints in Fig. 3 would thus be unchanged if we were to replace 10 by 1 in (17). We have added a note after (17) to clarify this point.

Referee 2

“I have read the paper with interest and have also looked through the communication with earlier referees. It utilizes recent results in MHD theory and combines them into a comprehensive assessment of the magnetic field levels that can be expected in the voids at the present time. As I have stated below, it is more of an assembly of recent results and applying them to cosmology rather than new per se.”

“I think the paper is basically publishable, but there are several points where the paper seems to claim more than is justified. Below my detailed comments in the order they appear.”

We thank the referee for their consideration of our manuscript and their comments — we are glad that they feel our manuscript is close to being publishable.

“1. In the abstract, I think it would be important to clarify when the authors are talking about nonhelical or helical fields. ”

This is a fair comment — we had, for reasons of brevity in the word-limited abstract, neglected this distinction on the grounds that generation of helical fields could be excluded at the EWPT for reasons of parity violation (we expanded on this in the main text). However, we take the referee’s point that not distinguishing the cases clearly could be misleading, and so have now clarified in our abstract that no parity violation is assumed (also restructuring its final sentence somewhat to satisfy the word limit).

“2. Regarding the lower limits, as already emphasized by an earlier referee, it would be important to spell out that those limit are subject to the assumption that plasma instabilities are not important. The recent review by Addazi+22 has some recent assessment on this. Since this is potentially a major threat, I suggest a mention this in several critical places in the paper. ”

We thank the referee for the suggestion. We have added three references to the possible effects of plasma instabilities: below Eq. (1), at the end of the section entitled “inconsistency with observations”, and below Eq. (20).

“3. The discussion of the magnetic version of the Loitsyansky integral under "Results" on p. 2-3 seems excessive and was never seen in MHD simulations. As the authors discuss only later, simulations have shown a slower decay, and the point of HS21 is that it clarifies the reason behind it. I think this should be reflected better in a revised presentation. Since the work of HS21 is fairly recent, any additional confirmation of their results are of course welcome. ”

We recognise that we are providing context of a review-kind in the highlighted discussion. However, we feel that this context is valuable as an explanation of why a theoretical treatment based on the “selective decay of small-scale structure” fails — because I_{L_M} is not invariant. This idea is presented in a number of prominent reviews on primordial magnetic fields, and new papers on the subject continue to mention the Loitsyansky integral and/or selective decay of small-scale structure (e.g., Alves Batista & Saveliev, 2021; Kahniashvili et al., 2022). We therefore felt that it would be valuable to give an account not only of our new theory, but also of why these old ideas do not hold.

We note in passing that early simulations did in fact support the old theory — see, for example, Fig. 2 of Banerjee & Jedamzik (2004), which appears to show preservation of the large-scale part of the spectrum. Reppin & Banerjee (2017) identified a diffusive numerical scheme as the cause for this. As our article elucidates, a more precise reason is that a diffusive numerical scheme does not conserve I_H — it might be said that magnetic field diffuses rather than reconnects under these conditions.

“Eq. (4) is only correct when $n=2, 6, etc.$ The authors may want to add a $-(-1)^{n/2}$ factor.”

We think there is a typo in this comment — most likely the referee refers to Eqs. (61) and (62). These equations are indeed missing such a factor, which we have now added — we thank the referee for spotting the error.

Referee 3

We thank the referee for their consideration.

“I am afraid that I cannot recommend the paper for publications,”

“First, I do not see enough justification to transfer the manuscript from [redacted] to Nature Communications. I do not see the novelty of the manuscript as well as importance for a wide audience to be published in Nature. ”

We are disappointed that the referee does not consider our article suitable for publication in *Nature Communications*. Our submission to the *Nature* family of journals is based on our belief that — in addition to being of practical importance both (i) to cosmologists seeking to constrain magnetogenesis models and (ii) to observational

efforts to constrain properties of magnetic fields in voids — our work will be of general interest to researchers of the wide variety of astrophysical systems for which magnetic reconnection and turbulent relaxation have key dynamical or observational consequences (e.g., the solar corona, active galactic nuclei, pulsars, etc.). We accept that the referee does not share our view — judgements of importance for a wide audience are after all in the eye of the beholder. On the other hand, we note that the stated aim of *Nature Communications* is to publish articles that “represent significant advances of significance to specialists”, rather than necessarily to report work that is of importance to a wide audience. That our article represents an important advance of this kind is particularly attested by the first report of referee 1, where they describe it as a “profound revision of understanding of evolution of primordial magnetic fields”. For concreteness, the main novel results of this article are:

- **The decay of primordial magnetic fields generated at a phase transition would respect the conservation of the Saffman helicity invariant, and not of other invariants (e.g., Loitsyansky) as has been supposed previously in the literature.** Although we acknowledge that we reported our discovery of the Saffman helicity invariant in an earlier publication (Hosking & Schekochihin, 2021), it should be noted that the earlier publication considered a general problem in turbulence theory without application to primordial magnetic fields (or to any other particular physical system). We think it is therefore unlikely that specialists in primordial magnetism would be aware of it. Indeed, we note that prominent recent work on primordial magnetic fields [including the paper by Kahniashvili *et al.* (2022) that the referee references later in their report] references the canonical (but, we believe, incorrect) theories of turbulent magnetohydrodynamic decay without mention of the one presented by Hosking & Schekochihin (2021). This comment is not meant as a criticism of such works, but as illustration of the importance of the present article, which (i) asserts the importance of the Saffman helicity invariant in the context of primordial magnetism and (ii) determines the consequences of its conservation for the expected strength of the relics of primordial fields that survive in cosmic voids.
- **The end point of the decay of primordial magnetic fields is not determined by the phenomenological “largest processed eddy” relation that has been employed in previous work, but instead by the physics of magnetic reconnection.** We stress that this is a significant revision of previously accepted theory — for example, in their seminal work on primordial magnetic fields, Banerjee & Jedamzik (2004) state that “the exceedingly large Prandtl numbers in the early Universe allow one to neglect dissipative effects due to finite conductivity [such as magnetic reconnection]” (p. 3).
- **We provide detailed analysis of the different magnetic-reconnection regimes possible in the early Universe.** In particular, we estimate the extent to which the large magnetic Prandtl number of the primordial plasma inhibits reconnection outflows, the effect of photon drag on reconnection inflows, and the possibility of violation of the MHD approximation by the development of a reconnection scale that is comparable to the microphysical plasma scales. We are unaware of any such analysis of magnetic reconnection in the early Universe in the existing literature. We find that accounting properly for these effects increases the expected field strength of primordial relics by several orders of magnitude compared to what would be expected under the “largest processed eddy” relation.
- **We find consistency between the constraints derived from γ -ray observations of TeV blazars on the strength of magnetic fields in cosmic voids and the relic-field hypothesis of their origin, based on the standard scenario of magnetogenesis at the electroweak phase transition without parity violation, followed by magnetically dominated decay.** This revises the conclusions of previous work (Wagstaff & Banerjee, 2016) that appeared to rule out this possibility.
- **We argue that, under our theory, decay is insensitive to the large-scale asymptotic of the energy spectrum (see Supplementary Information).** This is unlike in the canonical theory, where this asymptotic determines the decay laws, and remains true (up to a critical exponent) even when the spectrum becomes sufficiently shallow for invariants associated with the local fluctuation level of magnetic flux to become non-zero. Also in the Supplementary Information, we provide new numerical simulations of magnetohydrodynamic turbulence that decays without fluctuations in the magnetic helicity, demonstrating

that the turbulence redevelops helicity fluctuations via the dynamo effect and then decays subject to their conservation. We believe that both of these results will be of importance to specialists in primordial magnetism.

“Second, I disagree with authors that they replied to my previous comments. Especially when addressing the definition of the Alfvén speed: I suggest to re-address carefully the paper by Hu & White, 1997, <https://journals.aps.org/prd/abstract/10.1103/PhysRevD.56.596> in order to express coupling between baryons and photons. Importantly, the Alfvén speed in such a definition will be time dependent ($B^2 \propto 1/a^4$ and $\rho_b \propto 1/a^3$), see their Eq. 30 - and correspondingly it must be properly accounted when applying the comoving quantities for numerical simulations.”

We agree with the referee — the “comoving” Alfvén speed for motions at scale L grows once the mean free path of photons becomes less than L , as photons then no longer contribute to the inertia of the plasma. This effect would probably require a simulation with kinetic photons to capture correctly, which is beyond the scope of our article. However, as we already point out in the article [above Eq. (30)], we do not expect that the choice of whether photons or baryons provide the plasma’s inertia would affect any of our conclusions. This is because the densities of radiation and baryons are similar at recombination, when decay terminates.

“Third, I would like to highlight the magnetic field total energy density can be several order of magnitudes larger than it was estimated previously (and without the formalism described by the Authors) if we just account the decay laws and proper consideration of the Big Bang nucleosynthesis limits, See Fig. 1 of <https://journals.aps.org/prl/abstract/10.1103/PhysRevLett.128.221301> ”

The purpose of this paper is to show that the observational constraints can be satisfied in the standard context of magnetically dominated decay between genesis and recombination. It is true that there are other schemes to increase the field strength, although we note that the referenced paper does not appear to provide one (this is not a criticism — providing one does not seem to have been the paper’s goal). Instead, it charts the evolution of primordial magnetic fields under a number of different candidate decay laws, and these *do not* include the ones that we believe are correct for magnetically dominated decay. That the authors of this paper appear to have been unaware of the theory presented in our paper on decaying turbulence (Hosking & Schekochihin, 2021), despite publication of the latter preceding submission of the former, is illustration of the importance of our present article applying the formalism to primordial magnetic fields, in our opinion.

“I have some concerns about other parts of the manuscript: Eq. 1 - λ is the energy containing length scale, not a correlation length. In fact, it is assumed that the magnetic field is described as an homogeneous characterized by a λ -scale (i.e. monochromatic) field, and B_λ vs λ physically differs from the magnetic field spectral description. ”

This comment appears to be repeated later in the referee’s report, where they write:

“It is unclear physical interpretation of λ_B in their Eq. 1. Indeed, the lower limit of the magnetic field in Eq 1 is the effective magnetic field strength that corresponds to the homogeneous field strength that is mathematically determined by the monochromatic spectrum, i.e. in the momentum space the spectrum can be approximated by $\delta(k - k_B)$ with $k_B \sim \lambda_B^{-1}$ so it is not correct to identify as the correlation length as λ_B . ”

We therefore address these comments together.

The referee appears to be concerned about the validity of our equating the energy-containing scale k_{ec}^{-1} , usually defined as the peak of the magnetic-energy spectrum $M(k)$, with the correlation length (also called the integral scale), which is conventionally defined as

$$k_I^{-1} = \frac{\int_0^{\infty} dk k^{-1} M(k)}{\int_0^{\infty} dk M(k)}. \quad (1)$$

It is readily verified that $k_{ec} \sim k_I$ (up to a constant of order unity) if $M(k) \propto k^p$ with $p \notin [-1, 0]$ on either side of $k = k_{ec}$. This condition is satisfied for the “causal” fields usually considered to result from magnetogenesis at

a phase transition, which are the sort considered in our manuscript. The equivalence of the scale appearing in the observational constraint (1) and the correlation length of the magnetic field is a standard assumption in the literature for these kinds of fields.

On the other hand, $k_{ec} \sim k_I$ for the scale-invariant magnetic fields with $p = -1$ that could be generated by inflation. We do not consider these sorts of fields in our work. We do not believe that our formalism, as presented in this article, should apply to them, as they do not have a well-defined energy-containing scale. This is consistent with the study by Brandenburg & Kahniashvili (2017), which found different decay behaviour for scale-invariant fields. We have added a comment to the manuscript (after Eq. 6) to clarify that our theory does not describe scale-invariant fields.

“I also believe that the causality requirements for the scale-invariant magnetic field works properly (the field is “frozen-in” at super-horizon scales while the field is decaying in sub-horizon scales).”

We agree that a different scheme would be appropriate to describe a field whose correlation length was larger than the horizon scale. We emphasise that we do not consider scale-invariant fields in our study.

“Additional Comments”

“I do not think that the Authors replied to Comments by Referee 2:”

“Comment: I have now gone through the paper “Cosmic-void observations reconciled with primordial magnetogenesis” by Hosking and Schekhochihin. The authors combine their theory for inverse transfer in non-helical decaying magnetically dominated turbulence and a notion from previous authors (and themselves) that the time-scale for this could be the reconnection time. They argue that this allows one to explain magnetic fields in voids as arising from electroweak phase transitions. All the pieces of this argument are at present fairly speculative. There is considerable controversy whether the gamma-ray observations do indicate magnetic fields in the voids or can rather be explained as due to plasma instabilities (Broderick, Chang, Pfrommer, 2012, ApJ, 752, 22) and the large body of subsequent literature. The strength and correlation scale of magnetic fields generated during the EWPT is uncertain, even whether the phase transition can be first order (the reviews cited by the authors), and the theory of Hosking and Schekhochihin is still being scrutinized critically by the community. Moreover, even without invoking reconnection time scale as being the relevant one for the decay, there exist work which suggests that if all the above conditions are valid, the void magnetic fields can arise from the EWPT (Brandenburg et al, 2017, Phys. Rev. D 96, 123528), which does not agree with the work of Wagstaff and Banerjee, precisely because non helical decay was found to be slower than assumed by the later authors. Thus I find the current paper somewhat speculative, and also not novel enough nor of such immediate interest to a large astronomical community. Thus I am not able to recommend it for publication in [redacted].”

À propos the scrutiny of our work by the community that was mentioned in this quote from the first round of referees’ reports, we note that some results have now emerged and appear to confirm our theory (Zhou et al., 2022; Brandenburg, 2023; Brandenburg et al., 2023).

“The Authors Answer: We thank the referee for their consideration of our manuscript and their comments on it. We have now acknowledged the alternative view of Broderick et. al. (2012) in our manuscript. We note that our work differs from Brandenburg et. al. 2017 (which we have also now cited) in that (i) we provide a theoretical explanation for the inverse transfer via the Saffman helicity invariant, and (ii) include the role of the magnetic reconnection timescale. As such, our quantitative predictions are different from those of the earlier work – our theory allows EWPT fields generated with scale significantly smaller than the Hubble radius, for example.”

“The presence of the magnetic fields is still under debates: the Authors did not discuss in details arguments for the large-scale magnetic fields in voids; For example they do refer to Briderick et al (2012) but discard more recent studies,”

- *“E. Broderick, et al. “Missing Gamma-ray Halos and the Need for New Physics in the Gamma-ray Sky,” Astrophys. J., 868, 87, 2018.”*
- *“R. Alves Batista, et al. “The Impact of Plasma Instabilities on the Spectra of TeV Blazars,” Mon. Not. Roy. Astron. Soc., 489, 3836, 2019.”*
- *“R. Perry and Y. Lyubarsky, “The role of resonant plasma instabilities in the evolution of blazar induced pair beams,” Mon. Not. Roy. Astron. Soc., vol. 503, 2215, 2021”*

“In fact they do not compare the cosmological magnetic field approach vs. plasma instability in the context of the filling factor, see Ref. Dolag et. al. “Lower Limit on the Strength and Filling Factor of Extragalactic Magnetic Fields”, Astrophys. J. Lett. 727, L4 (2011). In addition, there is no discussion concerning the time-delay effect, see Ref. M. Ackermann et al., “The Search for Spatial Extension in High-latitude Sources Detected by the Fermi Large Area Telescope,” Astrophys. J. Suppl., 237, 32, 2018”

We interpret this comment as a request to provide additional context for our results. We have now done so, and explicitly commented below Eq. (1) on the possible role of plasma instabilities in scattering the electromagnetic cascade [and, further on in the text, caveated our comparisons with observational constraints with the possibility of modifications to Eq. (1) due to plasma instabilities]. We thank the referee for pointing out these references.

“The Saffman helicity invariant has been studied and discussed earlier in the literature, including works by the Authors, see Ref. D. N. Hosking and A. A. Schekochihin, “Reconnection Controlled Decay of Magnetohydrodynamic Turbulence and the Role of Invariants”, Phys. Rev. X 11, 041005 (2021).”

We can only repeat our previous reply to the referee in response to this comment: the discovery of the Saffman helicity invariant and role of magnetic reconnection in mediating the decay of magnetically dominated turbulence was indeed already reported in our earlier publication. Nonetheless, we believe the application of this theory to the early Universe, and, particularly, our detailed examination of the different reconnection regimes possible in the early Universe, to be a significant advance on the important problem of the evolution of primordial magnetic fields.

“Comment [from original Referee 2]: It is also surprising to say that these fields can explain cluster magnetism without dynamos, given the large body of work that shows dynamos to be essential to maintain cluster magnetic fields; perhaps the authors mean that these fields would provide a seed field?”

“The Authors Answer: The referee makes a good point that the phrasing of our comment was not clear. Our intended meaning was that, while dynamos are indeed essential for maintaining cluster magnetic fields, they may not be necessary for growth, as the “seed” field would already be strong enough (after compression during structure formation) to explain the observations. We have revised the manuscript to make this meaning clear.”

“The statement by the Authors is misleading: at large scales (voids) the magnetic fields are unchanged (“frozen-in”) during the structure formation, while in the galaxy scales the observed magnetic fields are determined by non-linear processes including dynamos. The strong enough seed magnetic fields might be ruled the Faraday rotation observations, see Ref. R. Banerjee and K. Jedamzik, “Evolution of Cosmic Magnetic Fields: From the Very Early Universe, to Recombination, to the Present”, Phys. Rev. D 70, 123003 (2004).”

We agree that the magnetic fields in voids are unchanged during structure formation. Our point is that the relic fields could feasibly be strong enough at recombination that — after the compression that occurs during structure formation — they would reproduce the observed field strength in galaxy clusters. Thus, the role of the dynamo effect in the intra-cluster medium would be solely as a maintainer of magnetic field, not as a generator, which we consider to be an interesting prospect worth highlighting. This is the same scenario as described in the paper by Banerjee and Jedamzik to which the referee refers [and also in Banerjee & Jedamzik (2003), which we have already cited]. It is of course true that the cluster field maintained by (saturated) dynamo need not (and, in all likelihood, will not) retain memory, or structure, of any primordial seed. We have added this point to our discussion below Eq. 20.

“I also think they did not address Referee 2 other comments.”

We are disappointed that the referee was not satisfied by other responses that we made to Referee 2. We would be glad to clarify our position and make any appropriate adjustments to our manuscript if the referee were to make their concerns explicit.

References

- Alves Batista, R. & Saveliev, A. 2021 The gamma-ray window to intergalactic magnetism. *Universe* **7**, 223.
- Banerjee, R. & Jedamzik, K. 2003 Are cluster magnetic fields primordial? *Phys. Rev. Lett.* **91**, 251301.
- Banerjee, R. & Jedamzik, K. 2004 Evolution of cosmic magnetic fields: From the very early Universe, to recombination, to the present. *Phys. Rev. D* **70**, 123003.
- Brandenburg, A. 2023 Hosking integral in non-helical Hall cascade. *J. Plasma Phys.* **89**, 175890101.
- Brandenburg, A. & Kahniashvili, T. 2017 Classes of hydrodynamic and magnetohydrodynamic turbulent decay. *Phys. Rev. Lett.* **118**, 055102.
- Brandenburg, A., Kamada, K. & Schober, J. 2023 Decay law of magnetic turbulence with helicity balanced by chiral fermions. *arXiv e-prints* p. arXiv:2302.00512.
- Hosking, D. N. & Schekochihin, A. A. 2021 Reconnection-controlled decay of magnetohydrodynamic turbulence and the role of invariants. *Phys. Rev. X* **11**, 041005.
- Kahniashvili, Tina, Clarke, Emma, Stepp, Jonathan & Brandenburg, Axel 2022 Big Bang Nucleosynthesis Limits and Relic Gravitational-Wave Detection Prospects. *Phys. Rev. Lett.* **128**, 221301.
- Reppin, J. & Banerjee, R. 2017 Nonhelical turbulence and the inverse transfer of energy: a parameter study. *Phys. Rev. E* **96**, 053105.
- Wagstaff, J. M. & Banerjee, R. 2016 Extragalactic magnetic fields unlikely generated at the electroweak phase transition. *J. Cosmol. Astropart. Phys.* **2016**, 002.
- Zhou, H., Sharma, R. & Brandenburg, A. 2022 Scaling of the Hosking integral in decaying magnetically dominated turbulence. *J. Plasma Phys.* **88**, 905880602.

REVIEWERS' COMMENTS

Reviewer #2 (Remarks to the Author):

I have examined the revisions in response to the suggestions made by myself and the other referees and feel that the paper can be published in its present form. To my mind, the material in the paper is of cross disciplinary interest and does therefore deserve publication in NatComm.

Incidentally, I wondered why Figure looks so distinctly unsharp, and what the black lines or streaks mean. If this figure is to be kept in this form, it might be adequate to represent it in smaller size (but then with slightly larger fonts on the axes), and possibly even at two different times.

Reviewer #3 (Remarks to the Author):

The Authors responded to my previous comments. I recommend the paper for publication.

+++Additional comments about Reviewer#1's previous concerns:

I believe that the Authors replied to Referee 1 comment.+++

Cosmic-void observations reconciled with primordial magnetogenesis – replies to referees

D. N. Hosking, A. A. Schekochihin

August 8, 2023

We are grateful to all of the referees for their comments on the manuscript and are delighted that they all consider the paper to be publishable in its present form. We address the final comment of Referee 2 below.

Referee 2

"I have examined the revisions in response to the suggestions made by myself and the other referees and feel that the paper can be published in its present form. To my mind, the material in the paper is of cross disciplinary interest and does therefore deserve publication in NatComm."

"Incidentally, I wondered why Figure looks so distinctly unsharp, and what the black lines or streaks mean. If this figure is to be kept in this form, it might be adequate to represent it in smaller size (but then with slightly larger fonts on the axes), and possibly even at two different times."

We take the referee's point that the figure might have been represented in too large a form in the previous version. We have reduced its size in the new version, which is also at higher resolution. We have also added colour bars to make clearer the meaning of the "black streaks" (whose meaning was already explained verbally in the caption). We choose not to present the figure at two different times, as this figure is not intended to convey evolution in time.